# "I became a person again": Social inclusion and participation experiences of Ethiopian women post-obstetric fistula surgical repair

Tibeb Zena Debele[1,2]*, Danielle Macdonald[3]°, Heather M. Aldersey[2]°,
Zelalem Mengistu[4]°, Dawit Gebeyehu Mekonnen[1,2], Beata Batorowicz[2]

1 Department of Clinical Midwifery, School of Midwifery, College of Medicine and Health Sciences, University of Gondar, Gondar, Ethiopia, 2 School of Rehabilitation Therapy, Queen's University, Kingston, Ontario, Canada, 3 School of Nursing, Queen's University, Kingston, Ontario, Canada, 4 Department of Obstetrics and Gynecology, University of Global Health and Equity, Kigali, Rwanda

° These authors contributed equally to this work.
* 19tzd@queensu.ca, zenatibeb@gmail.com

## Abstract

### Background

Childbirth-related mortality and morbidity affect many women globally, especially in low-income countries like Ethiopia. Obstetric fistula–a preventable condition mainly caused by prolonged and obstructed labor–can lead to physical, psychological, and social challenges, affecting women's social participation and inclusion.

### Objective

This study aims to understand women's social participation and inclusion experiences post-obstetric fistula surgery.

### Methods

This study is part of a larger research project investigating the social inclusion process of women who have had obstetric fistula surgery in Ethiopia. For this study, we conducted a qualitative exploration of women's experiences, guided by a constructivist grounded theory approach. Twenty-one women discharged from fistula treatment facilities following obstetric fistula surgery were interviewed using a semi-structured interview guide. Data was analyzed using Charmaz's inductive analysis approach, which involves an initial line-by-line coding followed by focused coding to identify the most significant codes. Subsequently, sub-themes and themes were developed from the focused codes.

### Result

The data analysis revealed four themes reflecting the women's experiences of social participation and inclusion. These are the experience of recovery and the journey toward social participation, participating in expected and meaningful activities, the continued challenge with a romantic relationship, and formal and informal support. Overall, the women who

**Funding:** Tibeb Debele received funding from the Mastercard Foundation and Queen's University dean's travel grant. the funders had no role in the study design, data collection, and analysis, the decision to publish, or preparation of the manuscript.

**Competing interests:** The authors have declared that no competing interests exist.

received fistula surgery reported positive life changes, especially regarding their physical well-being. However, they continued to face social challenges such as financial hardship, reproductive health problems, and issues with marriage and family life, which negatively impacted their social participation and inclusion experiences.

## Conclusion

While more research is needed, the findings of this study suggest that the social aspects of obstetric fistula are crucial for healthcare professionals to consider. Providing appropriate care and support to address unmet social relationship, employment, and childcare needs could enable women to lead fulfilling lives.

## Introduction

Globally, many women suffer from childbirth-related mortality and morbidity. According to the World Health Organization (WHO), for each maternal death, approximately 20–30 women develop injury and disability [1, 2]. Obstetric Fistula (OF) is an example of one such disability, one which results in adverse long-term outcomes. OF is a childbirth complication resulting in abnormal openings between the urethra, bladder, uterus, cervix, ureter, and rectum, leading to continuous leakage of urine, feces, and flatus [3]. The United Nations (UN) reports that 500,000 women live with OF globally, and in Ethiopia, 0.6% of all pregnancies end in fistula, according to a national demographic survey [4, 5].

We used the International Classification of Functioning, Disability, and Health (ICF) framework, which conceptualizes health as encompassing physical and social well-being, to understand women's experiences of social participation and inclusion [6]. The ICF framework also helps understand how environmental factors, such as the community's attitude towards the cause and consequence of fistula [7, 8], may impact women's social participation.

Social participation–crucial for overall well-being and positive social behavior–is a key aim of rehabilitation [9–11]. The WHO (2002) defines participation as "involvement in life situations," and it encompasses learning, communication, mobility, self-care, relationships, community, and civic life [6]. Hammel et al. [2008] characterize social participation as active and meaningful engagement that involves choice and control, access to opportunities, responsibility, impact, social connectedness, social inclusion, and membership [12]. Encouraging social participation among marginalized groups, such as women with OF, can boost their well-being, quality of life, and social inclusion. The United Nations defines social inclusion as improving societal participation terms for disadvantaged individuals based on age, gender, or disability [13]. Social participation, closely linked to social inclusion, is a crucial consideration for marginalized people, such as women with OF, in alignment with the UN's priority for inclusion.

The treatment for OF involves surgical correction, and successful treatment is determined by the level of women's continence status [14, 15]. However, such measurements fail to address the psychological or social well-being of the affected women. Despite efforts to improve the social participation and inclusion experiences of people with various health conditions, there remains a lack of research on women's experiences after OF surgery. Therefore, this study aims to explore women's experiences with social inclusion and participation after OF surgery in Ethiopia.

## Materials and methods

### Research design

We conducted a qualitative exploration of women's participation and social inclusion experiences, guided by a constructivist grounded theory approach [16].This study is part of a larger grounded theory research project looking at the social inclusion process of women who have had OF in Ethiopia.

### Study participants and recruitment

The research involved women who had obstetric fistula surgically repaired at the University of Gondar General Specialized Hospital and Bahirdar Hamlin Fistula Hospital in Ethiopia and were discharged. Data collection took place between July 7, 2022, and November 30, 2022. Participants were women aged 18 or older, at least six months post-discharge, regardless of surgery outcome. Women were purposefully selected using maximum variation criteria, considering factors such as parity, stillbirth presence, and continence status.

Research assistants from the fistula treatment facilities assisted participant recruitment by contacting the women first. After obtaining verbal consent, the first author (TD) provided detailed information, answered questions, and sought written informed consent from each participant. Twenty-one participants were recruited until data saturation was reached. [17]. Data collection and analysis were undertaken simultaneously through an iterative process following the grounded theory approach of Charmaz [30]. Data started to repeat at the 19[th] participant, and we continued until the 21[st] participant when the research team agreed that saturation had been achieved as no new concept had emerged. The 21 participants were from various districts in the Amhara region, with most having vesicovaginal fistula. Over half had experienced stillbirth, and two-thirds were fully continent at the time of the interview. Participant characteristics are detailed in Table 1.

### Data collection

The interviews were conducted in Amharic [the local language] and followed a semi-structured interview guide developed for this study, as seen in Table 2. All interviews were done in person; 16 were conducted in the morning, and the remaining five were done in the afternoon. The interviews lasted 36–90 minutes and were audio-recorded. Interviews were conducted in participants' homes, local churches, and healthcare facilities. To ensure privacy, home interviews were held in separate rooms or rescheduled if others were present. For interviews at churches or public institutions, we chose weekdays when the locations were empty or coordinated with administrators to secure private spaces.

### Data analysis

We applied Charmaz's inductive data analysis approach [18] and used Nvivo version 14 software to manage and organize the data. Initially, each interview was transcribed verbatim in Amharic. The first author read the Amharic transcripts. Next, TD and a second coder (DG), both fluent in Amharic, did line-by-line coding in Amharic–generating as many inductive ideas as possible [16]. Afterward, focused coding–identifying the most significant codes was completed by the two coders. Finally, the two coders and the senior author (BB) developed the themes and subthemes, which were further refined by the research team. To ensure rigor, a language expert conducted the forward translation of codes, sub-themes, and themes, followed by backward translation by two experts proficient in Amharic, English, and with a health sciences background. During analysis, the research team compared data, codes, and incidents.

**Table 1. Characteristics of the study participants.**

| Characteristics | N | % |
|---|---|---|
| **Age** | | |
| 20–24 | 3 | 14.3 |
| 25–29 | 7 | 33.3 |
| 30–34 | 5 | 23.8 |
| 35–39 | 2 | 9.5 |
| 40–50 | 4 | 19.1 |
| **Marital status** | | |
| Married | 10 | 47.7 |
| Divorced/Separated | 11 | 52.3 |
| **Current Residence** | | |
| Urban | 7 | 33.3 |
| Rural | 14 | 66.7 |
| **Type of fistula** | | |
| Vesicovaginal fistula | 10 | 47.7 |
| Rectovaginal fistula | 9 | 42.8 |
| Combination of Rectovaginal fistula and Vesicovaginal fistula | 2 | 9.5 |
| **Number of years after developing fistula** | | |
| ≤4 | 12 | 57.1 |
| 5–9 | 1 | 4.8 |
| 10+ | 8 | 38.1 |
| **Number of years after treatment** | | |
| 6 Months -1 year | 5 | 23.8 |
| 1–4 year | 9 | 42.8 |
| 5–9 | 4 | 19.1 |
| 10 | 3 | 14.3 |
| **Number of surgeries** | | |
| 1 | 10 | 47.7 |
| ≥2 | 11 | 52.3 |
| **Outcome of indexed child** | | |
| Stillbirth | 11 | 52.3 |
| Alive | 10 | 47.6 |
| **Continent status** | | |
| Incontinent | 5 | 23.8 |
| Fully continent | 14 | 66.7 |
| Semi-continent | 2 | 9.5 |

The first author prepared a reflexivity statement before the study to examine potential biases and maintained a reflexive journal throughout, discussing reflections with the PhD supervisor and the research team as needed [19].

## Ethical clearance

We obtained ethical clearance from the Health Sciences and Affiliated Teaching Hospital Research Ethics Board of Queen's University (Ref.No-6036424) and the institutional Ethical Board of the University of Gondar (Ref. No-VP/RTT/05/1004/2022) to commence this research. Written informed consent was taken from each study participant.

**Table 2. Semi-structured interview guide.**

| S.N | Questions |
|---|---|
| 1 | Can you please tell me about the health condition you have? |
| 2 | When you first learned you had a fistula, with whom did you share the information about your condition and why? |
| 3 | How has developing OF changed your relationship with others [husband, parents, siblings, friends] and things you could do? Probe: What reactions did OF bring to your family? |
| 4 | What medical and rehabilitation services were you given after attending the fistula institutions? |
| 5 | After you had surgery and completed your medical treatment, where did you go? |
| 6 | How have things been for you since you had surgery and returned from the treatment facility? |
| 7 | Please describe your daily activities and participation in your community since you returned. |
| 8 | How has your relationship with your partner/spouse been since you returned to the community? |
| 9 | How did people in your family and community perceive and treat you after you return? |
| 10 | What kind of support have you received since you returned that has helped you settle into the community? |
| 11 | What support did the community give you that was helpful to your inclusion and participation? |

## Results

Our data analysis resulted in four main themes that reflect women's lived experiences regarding their social participation and inclusion post-OF surgery: (1) The experience of recovery and journey toward social participation; (2) Participating in expected and meaningful activities; (3) The continued challenge with romantic relationships and (4) Formal and informal supports. The themes and sub-themes are presented in (Table 3).

### The experience of recovery and journey toward social participation

This theme consists of three sub-themes: living with fear, being resilient and hopeful, and developing strategies to participate, which reflect how women experienced their recovery post-OF surgery in relation to their social participation and inclusion. Women discussed what caused them to participate or not in their social circles (families, neighbors, work colleagues, and the community at their church), along with the strategies they used to enhance their participation and social connections.

**Table 3. Themes and sub-themes.**

| Themes | Sub-themes |
|---|---|
| The experience of recovery and journey toward social participation | Living with fear |
| | Being resilient and hopeful |
| | Developing strategies to participate |
| Participating in expected and meaningful activities | Reconnecting with close people and the community |
| | Engaging in self-care, household, and community activities |
| | Employment and meaningful work |
| The continued challenge with a romantic relationship | Choosing to stay single |
| | Worrying about the sustainability of marriage |
| Formal and informal supports | Having positive family support |
| | Support from community groups and organizations |
| | Need for rehabilitation services |

## Living with fear

Participants shared how they suffered from the continued effect of OF post-surgery and rehabilitation due to their traumatic incident. Even the women who were fully continent following the surgery kept worrying about the risk of the fistula reoccurrence or fear of incontinence in public places. Participants highlighted that they feared sexual activity, giving birth to another child, strenuous work, and travel could trigger fistula occurrence. For participants, the fear of reoccurrence not only caused psychological stress but also restricted their outings and social and community participation. One participant said:

> Now, I am not as I was before. I was fast and quick before. In the past, I went from Woreda to ወረዳ [district to district] and brought things [to sell]. Now, I am getting too weak to travel. And I am afraid, suspecting the stitch may be undone when I move. (p-06)

The experiences of fear resulting from OF affected various aspects of women's lives. One woman who divorced twice expressed the fear of raising her children alone: "Divorce is something severe; marriage is what I like. Death is [would be] better than separating after giving birth to a child. Raising children, being a single parent, without a husband is difficult." (p-01) Another woman who was cured of the fistula and was actively working highlighted how she worries about her future: "I think to the future when I work day in and day out, I see it coming, this [fistula] can happen again. When my physical condition starts getting weaker and weaker, maybe it [fistula] might happen." (p-17)

## Being resilient and hopeful

Some women who were not cured of the fistula after surgery accepted their condition as a lifelong problem that cannot be solved. Those women found that accepting their condition helped them focus on improving their daily lives, while others showed an optimistic attitude about the possibility of getting cured. One participant who had two previous surgeries and was waiting for her subsequent surgery said:

> They told me it would heal when surgically treated. Even though it was treated first and now again, it didn't heal, even if they promised [reassured], it relapsed and is as it was. They gave me an appointment to come [fistula facilities] in six months. . . I am waiting for the appointment day. I think that may bring some hope. (p-03)

Women who underwent successful fistula surgery overcame shame and guilt and experienced a positive shift in self-image, and started setting bigger life goals. One woman explained:

> Prior to having the treatment [surgery], your life is shameful. It seems to one that it is not possible to live equally with others; it seems one may be left out. One may choose to live alone if not treated; after treatment [surgery], you feel you are equal and a full person. (p-08)

## Developing strategies to participate

Some women accepted their condition, believing it was a God-given problem. One participant said: "God gave us this problem [fistula] once. It is not man-made; yes, it is God-given. Some people have experienced the problem, then they get cured and normal, but we, the unlucky ones, are still like this." (p-07)

Almost all women viewed their fistula as a challenge they were destined to overcome; returning to their "previous states" meant passing a divine test. Such spirituality and faith served as a source of strength. One woman explained:

The saying, 'God tests the ones he loves the most,' came true to me. I used to think it [fistula] only happened to me, not knowing it was meant to give me a lesson. I complained about why God tempts me; what severe sin did I do [great sin did I commit]? However, when I pass all these tests and see myself living a peaceful life with my children, I am happier now; I am happier than before. (p-16)

Women who were not cured of the fistula used strategies to increase social inclusion, such as avoiding eating and drinking in public, timing their interaction with others, trying to 'look normal', or 'telling others they are cured of the fistula'. One participant highlighted:

When I find people like you [educated], I tell them there might be a solution. However, to other individuals, I say I am cured. However, if you or the doctors ask me, I feel they might give me a solution and, therefore, tell them [about my fistula]. To others, I do not say that I am not cured. (p-19)

## Participating in expected and meaningful activities

Discussion with women post-surgery revealed two key sub-themes regarding their participation experiences. Firstly, they shared positive experiences and challenges in reconnecting with family and community. Secondly, they noted increased independence in daily activities post-correction. Lastly, economic hardship remained a concern, with limited employment opportunities hindering social participation and inclusion.

### Re-connecting with close people and the community

Women discussed reconnecting with family, neighbors, and community members post-surgery, noting the positive impact on their emotional well-being. One participant who was cured of OF expressed:

We have weddings and funerals; thank God I also participate in Christianity. I participate and attend such events, contributing payments equally as others do. I contribute as much as others. Thanks to my God, I am well and do as people do. (p-12)

Even those women who were not cured of the fistula reported positive experiences of meeting with family and friends. One woman said:

They [family] are fine now, even if it is not like before [relationship with family], they [treat me better than before]. They invite me to ማህበር [an association], and they call me for a coffee. Those who are closer invited me. Even those who are far they invite me when holidays happen. And I am not as intimidated now as I was before. I feel at ease now when I go to them. (p-13)

Some participants mentioned avoiding social gatherings. Those who had successful surgical correction avoided social gatherings for reasons such as lingering physical pain and those with uncured fistula because of stigma and social disconnect. One woman, still uncured and residing in the community, highlighted:

No, the community does not welcome me; I am outcasted. To tell you the truth, I am highly outcasted. Had the community accepted me, would I sit like this [in a filthy space]? You become happy when people console and approach you. (p-04)

The study participants highlighted how their surgery and the other supports they received limited their previous social isolation and positively affected their emotional state. One participant mentioned:

I was saying and wishing to be cured for one month and die afterward. Being together with other people is being able to do that now [gathering]; I cannot express the degree of my pleasure in words. The reason is that being isolated from people is the heaviest thing one can experience in life. (p-08)

Some participants mentioned re-engaging socially immediately upon returning home after being cured of the fistula. Those participants who returned to their social circles highlighted how reconnecting with others improved their overall well-being across various aspects of life, such as their physical appearance and sleep patterns: "I feel better. I sleep; I have no worries. I am fine now" (p. 17). For many, being cured of the leakage allowed them to participate in meaningful and valued religious ceremonies with their families and community. One participant highlighted: "When I kneeled [for prayer], there was leakage of gas, and because of that, I was very embarrassed. Now, Alhamdulillah, I can bow down together with other people." (p-01)

## Engaging in self-care, household, and community activities

The women reported notable improvements in household tasks after undergoing OF treatment, including cooking, cleaning, and childcare. One participant said: "I mop my house, cook, and feed the child, for him [husband] when he goes to work, I cook for him as much as I can. I wash clothes, I do all these things." (p-12). On the other hand, one woman emphasized enduring pain that interfered with her daily activities:

After that occurrence [fistula], I feel it [pain] very much, I get sick when I travel. . .there are many problems. I did not do house chores, nor did I travel easily. When I do not attend funerals, people say, 'What is wrong with her? it is near; she can go to a place called Robit [nearby place]?' But, I say, 'Let them say whatever they may, I do not worry.' (p-20)

Most women who underwent successful fistula repair noted improvements in maintaining hygiene post-treatment: "I am fine now; in the past [during the fistula], I used to ask [myself] why would I wash? But now I wash my clothes and live a relaxed life." (p-13)

## Employment and meaningful work

Participants discussed how fistula significantly affected their economic situation due to their inability to work, leading to financial struggles that persisted post-surgery and impacted their families. One participant noted:

We used to have cattle, but we sold them to cover the expenses for treatment and the holy water. The remaining small things were sold, and the [money] was sent to us; other than that, we had nothing. Our house is a rural house. It cannot be sold. (p-12)

Both groups of women (those who had successful OF surgery and those who continue to experience involuntary leakage) mentioned that their fistula had caused financial hardship. This hardship continued to affect them since they could not find work post-surgery. Economic hardship and struggle to find work post-surgery were common characteristics across cured or uncured women. One participant who had an unsuccessful surgical repair and who in the past had her own small business explained:

> People talked about me, and when people talked about this [my health background], they stopped buying bread from my shop, and the shop finally went bankrupt. When you are like this [fistula], people are [hateful]. When you try to work, they say she is like this [gossiping about the health background]. As a result, customers shift to other shops, and my life is a misery [crying]. (p-04)

Some participants noted improved finances post-surgery, taking the initiative by starting their own businesses. One participant stated: "Well, I borrowed some money from people, and I started selling tea, then I bought materials for sewing with the money I got from selling tea. Then I repaid my debts." (p-13)

## The continued challenge with romantic relationships

This theme captures women's post-OF surgery experiences in marriage and romantic relationships with two sub-themes: (1) choosing to stay single and (2) worrying about the sustainability of marriage due to ongoing issues.

### Choosing to stay single

Over half of participants reported divorce or separation due to the negative impact of fistula on their marriages, which persisted post-surgery and rehabilitation. One woman discussed her divorce experience as follows:

> I caught the disease because of him [I married him]. He left, leaving me alone. He left me immediately when he saw me sick. He left me within two days when he saw me 'ተበላሻሽቺ' [messed up] [feeling overwhelmed by the leakage]. When he saw me in that state, he immediately judged that I was no longer of use to him. Then, he disappeared. (p-04)

Participants who were single, divorced, or abandoned expressed a desire to remain single even after undergoing surgery. Their experiences of betrayal and abandonment led them to question the essence of marriage, and they developed trust issues towards significant others. One woman whose husband left her and started a new life elsewhere stated:

> I have decided not to remarry because of the refusal and betrayal of my husband, the father of my children, left me in a time of trouble. Also, how can I get married while having this [fistula]? So, I have taken it as a God-given grace. . .I am considering taking the holy communion and following the order [restricting any sexual activity]. I do not have any thoughts of marriage. (p-07)

Only one woman justified her husband's decision to divorce her, believing that healthy individuals shouldn't suffer alongside those with health issues: "He came, visited me, and left, would a man who is working stay with a woman who is ድዋ [cursed and disabled]?" (p-05)

## Worrying about the sustainability of marriage

Fertility issues were also factors that women mentioned as the cause for their decision to stay single despite being cured of the fistula. The women described how their inability to conceive a child affected them and their family members. Women felt they were not good enough if they could not fulfill their husband's desire to have children or felt their children would not have a sibling. Therefore, they believed staying single was the best decision. Also, after surgery and rehabilitation, women faced criticism and pressure from extended family members urging them to end their marriage due to infertility. One woman cited infertility as the reason for divorcing her husband after fistula correction, aiming for her child to have a half-sibling from their father:

> I repeatedly insisted him remarry. I said to him marry and beget children for our child to have a brother or a sister. I said to him, 'No problem.' If you like, I will search for a woman to be engaged with you as a wife. We separated, having this agreement. (p-02)

For some participants, continued fistula despite surgery and rehabilitation was another reason for choosing a single life. Women highlighted how the leakage of urine, feces, and flatus makes it challenging to engage in marital life. One woman who was still incontinent despite two previous surgeries stated:

> No, I haven't tried it [a romantic relationship] yet. I think it is because of the fistula. Well, how can I have any relation with such a partner, being in this state [fistula]? I know it is necessary to be faithful to one person; the problem is the question of how being in this state so far, I have no plan for a romantic relationship. (p-03)

On the other hand, one woman shared her desire to rebuild her marriage with her former husband now that she had regained her health. Emphasizing the value of a life partner, she noted:

> What do I ask God? That guy [previous husband] told me he will be back with me if I get cured. If I get back with him, for one thing, we have a similar case [HIV]; second, when you are a couple, there is no suffering. . .you do not desire another person; your home will also be full [economically provided for]. (p-20)

Women spoke about the implication of their divorce on their social inclusion. One woman whose marriage ended noted her mother's lack of acceptance of her condition and divorce: "She [my mother] was not happy. She said, 'My daughter, you are sick and divorced.' She was not happy; she was not receptive [of me]." (p-18)

Many married participants voiced concerns about the future of their marriages, particularly regarding intimacy. They felt unable to fulfill their role as wives due to pain during intercourse, fearing it could lead to divorce. One participant stated:

> It is obvious, as I told you, that if you do not meet/fulfill [the sexual] desire of men, their hidden emotions get aggressive. He [husband] is not happy. I heard him blaming me because I did not fulfill his desire. He [threatens] me, saying, 'From now on, I will search for another woman. I was not sure whether it was true or simply to scare me. For my part, I was saying, 'If I cannot [perform sexual activity], you can search for another woman.' (p-10)

Fear of fistula reoccurrence also reduced their activity levels. One participant said:

When I am around, he stops talking, saying, 'Unless I have a child, I cannot live together.' Most of the time, he is silent. If he desires to marry another woman, I will divorce and divide our property, and he would prefer to live independently from today onwards. His family also loves children and asks, 'What will she do for you if she does not give birth to a child.' (p-17)

On the other hand, some women reported enhanced relationships with their husbands after surgery, citing improved communication, understanding, and intimacy. Some even mentioned conceiving and having children post-surgery. One pregnant woman elaborated:

Nowadays, marriage is very difficult. I am very happy when I have been told I can give birth to up to three children. Right now, I am pregnant. Hmm, if I am told I cannot give birth, and if he[husband] told me he would marry a woman who could give birth to a child, what would I feel? It would only be death that would free me from the suffering. (p-11)

## Formal and informal supports

This theme consists of three subthemes: (1) having positive family support, (2) support from community groups and organizations, and (3) needs for rehabilitation services. This theme talked about the positive experiences as well as areas women need more assistance with to enhance their social inclusion and participation experiences.

### Having positive family support

Some participants noted positive support from their family post-surgery. They mentioned that family members were welcoming upon their return home and provided emotional and moral support to re-engage in the community and meaningful activities. One participant indicated: "They encouraged me to start my own business so that I can be employed. When I told them I was cured and staying at home doing nothing, they advised me to start my own business." (p-15).

Women received in-kind support such as nutritious food provision and grain aid from their family members. Women also received physical assistance from family members, particularly from other women. They noted how these individuals helped with childcare, household chores, and personal care as needed. One woman said:

". . .She [mother-in-law] is the one who bakes injera [local bread] for me. There is a woman who grew up in her [mother-in-law's] house, and she told the woman to help me, informing her that I have a health problem. Even though she [mother-in-law] is not capable of helping me, she usually tells that woman to support me. She even tells her to wash the children, explaining that I need help. (p-16)Women emphasized the positive impact of close family support, which motivated them to persevere during periods of poor health when they couldn't engage with others. Some utilized this support to maintain social participation even when physically unable to do so. One participant explained: "Before I became sick when I was invited, I used to send them injera [local bread], and I also used to go there. However, now, I only send the little child; I do not go." (p-20)

However, a few women reported lacking family support from the beginning. One participant felt abandoned by her family: "Nowadays, no one calls to me. I live by getting strong and

encouraging myself." (p-04). For other women, even if they initially had family support, it dwindled over time: ". . . I do have brothers and sisters; everyone worries about themselves. Nobody cared when I had my second surgery; the first and the second [surgery] was not similar [the support from family]." (p-11)

Participants stressed the importance of social support from friends, relatives, and organizations. One woman highlighted how her mental well-being could improve with help from people in her social circle: "When someone comes and exchanges ideas [words of hope] saying 'you will be better soon,' whether it is a friend or a relative, your mindset will be renewed." (p-04)

## Support from community groups and organizations

A few women mentioned receiving support from neighbors and informal social groups, including visiting and providing clothing. This support fostered a sense of belonging to the community. One woman noted how her female association members supported her in staying in the social group despite her inability to pay the membership fees. She said:

> I did belong to a social association called 'እድር [Edir], but then I got sick. Even though my name was found in the list of the 'Edir,' I did not pay the monthly contribution for ten months; they [members of the Edir] have asked me if I wanted to leave or stay in the association. I told them I did not want to be canceled from the Edir, but I could not pay the payment timely. I asked them if they could accommodate me and give me time . . .later I paid the payment, and I am still in the Edir. (p-06)

However, most participants felt that their community didn't offer the support they hoped for and were unreceptive to their needs. One participant mentioned opting not to disclose her health condition, potentially leading to a lack of support from the community: "For the community to give me support, they do not know about my condition. Only my family knew about my condition. They [the community] have only heard I have had surgery recently." (p-17)

Few participants highlighted society's negative and stigmatizing attitude towards them and other women with OF. They felt the community was unsupportive and unwelcoming. One woman with a fistula, despite two surgeries, expressed her struggle to settle in any house due to being frequently evicted from rental houses. She indicated:

> They say that there is a bad odor. I left many rental houses, for they said there was a bad odor because of my presence. Including this house [a house she previously lived in], I left 12 others. Now again, I hear a rumor about my expulsion [current rental]. They said they were suffering from flu-like symptoms from the smell. . .I have been here for six months, but I do not have any other option but to leave. (p-04)

## Need for rehabilitation services

Women discussed positive experiences in their treatment journey, including rehabilitation services like training about fistula causes and prevention, counseling for psychological and social consequences of OF, and receiving items like clothes and mobile phones. However, they mentioned unmet needs for meaningful employment, support with sexual life challenges, and access to community-based services.

Participants spoke positively about their experiences at fistula institutions where they received surgical repair and rehabilitation services. They appreciated the counseling, skills

training, and grants provided. Regardless of surgery outcomes, all participants praised the welcoming approach of the institutes and how the services helped them regain their identity. One participant mentioned:

> There is the treatment [surgery], they provided us food when I was admitted at the hospital after treatment, they followed and took care of me. They [people working at the fistula center] are not any different from a father or a mother; their care is special. They cared for me, and I became a person again because of them; as you can see, I can walk again. (p-19)

Women felt respected and valued at fistula institutions, which positively impacted their emotional well-being. One participant said:

> When you compare workers at the fistula institutions and other places, they are far apart, like the earth and the sky. Wherever you are [in the fistula institute], you are not called for being wrong; at the fistula institute, they give you respect, they give respect for their patients, and all this should make you pleased. (p-22)

In addition to professional support at the fistula institutes, women also found support from peers facing similar health challenges. They were motivated by others' experiences to persevere. One woman explained:

> When they told me, 'You are lucky you identified fistula treatment within a short time.' Some people lived with the fistula for three or four years. I was consoled. I would have been dead by now. Once I came here and got admitted, I saw my friends [other admitted women] when they told me they lived with fistula for a certain year and said, 'You are fine; I got better.'(p-12)

Participants received various rehabilitation services from an NGO, including short-term education, small grants for starting a job, clothing, nutritious food support, counseling, and communication equipment: "We trained for two weeks, they gave us a radio and money, which amount I do not remember. I was given clothing and a phone. After being given that, we graduated [selected as an ambassador to advocate for fistula treatment and prevention]." [p-09] While most participants didn't receive support upon leaving fistula institutions to reintegrate into family and community life, a few did receive aid like grains and sanitation equipment. This assistance helped them feed their families, start businesses, and regain independence.

Despite positive experiences at fistula centers, participants expressed unmet post-surgery needs and desired more rehabilitation services. Many emphasized the necessity for support to return to paid employment. One participant felt the assistance provided wasn't sufficient, especially considering current inflation:

> I mean, the amount [money] given [at the rehabilitation center] is enough to start selling tea and coffee; however, when you see the current situation, everything is costly, and there is also [house] rent. Though it is good [the support] for me, what I think is I may start working by taking additional [financial support] from my family. (p-15)

This example highlights women's challenges when starting a business after OF surgery. The support provided was insufficient for sustainable livelihoods, leading women to rely on their families for financial stability. Many participants questioned the usefulness and sustainability

of the available minimal financial aid. One woman mentioned a well-established organization that was formed to support the vocation of women post-OF surgery had come to an end for unknown reasons and with no future plan for women whose livelihood was dependent on it: "We [uncured fistula patients] were trained and brought together for weaving thread, and seamstress and had a house built for us for three years, then they [funders] demolish it, yes, they demolish it." (p-05)

Most women still with their husbands lacked support for sexual well-being. One woman said: "They [fistula institutes] should give counseling bringing husband and wife together about the general condition that exists [about sexual relationships] including the safety measures they must take." (p-10)

Women with unsuccessful surgical outcomes discussed the need for additional procedures to address urine leakage. One woman, living with a fistula for a decade, emphasized the priority of being cured before any support could be meaningful.

> If you are not clean and healthy, people might not approach and chat with you, nor will they buy anything from you. Therefore, you go bankrupt. . .Now, I am thinking about how to get treatment once and for all and get my health back. (p-04)

While prioritizing fistula treatment, many women stressed that additional support in their daily lives could improve social participation and inclusion if the condition couldn't be resolved. Apart from a minority receiving local financial aid, most participants lacked community-based support services.

## Discussion

Despite the positive impact of fistula surgery and support, women still faced challenges in various aspects of their lives, including social, psychological, sexual, and vocational well-being. Marriage, family, spirituality, community life, employment, and support services were crucial for inclusion and participation. Their experiences underscore the importance of fistula treatment facilities, peer support, and services addressing marriage and family support. Gaps in informal support were also noted.

### The transformative power of fistula treatment facilities and peer support

Women attributed positive experiences at fistula institutions to the respectful attitude of health care providers and support from other women with OF. Our study participants faced abandonment, stigmatization, and exclusion from their families and communities, similar to the experiences of other people with other disabilities who, as a result, developed issues with self-esteem, belonging, and lack of dignity [20–23]. The experiences at the fistula treatment facilities enabled them to heal from the physical trauma and regain their dignity. Similarly, a grounded theory study by Williams and Irurta among hospitalized patients in Western Australia linked dignified health care and emotional comfort with a positive outlook toward life, improved mood, and fast recovery [24].

A notable finding from our study is the value of informal peer support among women undergoing fistula treatment. Participants highlighted how emotional support and companionship from peers facing similar health issues helped alleviate feelings of shame, guilt, and anxiety. Research suggests that the social comparison theory [25] can explain health-related issues, as comparing one's health with similar others can normalize experiences, boost self-esteem, and provide positive role models [26–28]. Strengthening postoperative informal peer-led support could potentially enhance women's social participation and inclusion experiences

after OF surgical correction by boosting their self-confidence and facilitating the normalization of their traumatic experience.

## From surgery to self-sufficiency: The need to support obstetric fistula survivors

Participants' narratives revealed unmet needs post-OF surgery. Despite expressing optimism about the services offered at the fistula centers and rehabilitation facilities, women reported a lack of economic support and services for their sexual health. A UN report on fistula highlights the importance of a holistic approach to addressing the psychological and socioeconomic challenges faced by OF survivors [4]. Our study findings indicated that some women who had OF surgery received rehabilitation services like counseling and life skill training, while others did not. Furthermore, despite the services provided, many women still struggled to find income-generating employment due to a lack of adequate financial support. Financial aid, if available, often failed to support starting or maintaining small businesses. In addition, unsupported women faced extreme poverty and lived vagrant lifestyles. Prior studies have also noted similar economic challenges post-OF surgery [29–31]. For example, a Ghanaian study found that women often faced impoverishment upon returning home after OF surgery, having depleted their life savings to seek treatment [32]. Meanwhile, a Ugandan study highlighted economic challenges post-OF surgery and proposed solutions such as microfinancing, grants, and community economic forums for the successful return of women to their community [33]. Our findings underscore the need for meaningful economic aid for women post-OF surgery to enhance their employment participation. However, there's a notable absence of practical solutions to improve their economic capacity, highlighting the need for further research.

Our study highlights a significant lack of community-based support programs for women post-OF surgery. While a few participants of our study received non-financial aid from local fistula rehabilitation centers, the majority reported insufficient support. A study in Tanzania echoes these findings, emphasizing the need for counseling services within the community once women are discharged from fistula treatment facilities [34].

Community-Based Rehabilitation (CBR) programs have a positive impact on the lives of individuals with disabilities [35], including improved functioning through better family support, enhanced income, and self-esteem [36], improved social participation [37], and better emotional well-being [38]. Women with obstetric fistula face one of the most debilitating maternal health conditions, often feeling disconnected and unsupported by their families and communities. CBR could address their skills development, self-employment, relationships, and family life needs. However, the availability and effectiveness of CBR programs for these women are largely unknown. Further research is needed to explore CBR's potential in providing sustainable support to women with OF patients.

## Marriage and family

In our study, most women were divorced, which is consistent with previous OF research in Nigeria, Ghana, and Ethiopia [39–41]. While most divorces in our study occurred before surgical intervention, aligning with prior OF studies in Ethiopia and Uganda, in some cases, divorce followed successful surgical correction, not necessarily due to OF but rather other sexual and reproductive health issues, such as infertility [34, 41, 42]. Researchers in Tanzania and Malawi have documented ongoing sexual and reproductive health issues post-OF surgery, including infertility and sexual dysfunction [40, 43]. In numerous African countries, marriage and childbearing are crucial for women to access social and economic rights [44–47]. Disruptions to the marriage structure can lead to various consequences for women, including

poverty, single parenthood, social isolation, challenges in forming new relationships, and stigmatization, as noted in broader literature beyond OF [48–53]. While our study and others have highlighted high rates of divorce among women with OF, research on the significance of divorce post-surgical correction and its impact on social inclusion is lacking. Further investigation is needed to address this important social issue.

Our findings revealed ongoing concerns about sexual health among women, contributing to marital tension [31]. A study in Kenya by Khisa similarly found that post-surgical abstinence and loss of the uterus affected marital stability post-OF repair. Additionally, a study from Guinea highlighted the importance of successful reintegration, including resuming marital life and engaging in sexual activity, which is often not achieved [54]. Research on divorced couples outside of OF has shown that sexual problems can impact marital stability [55, 56]. Addressing sexual issues and providing accessible platforms for women to voice their challenges could lead to better long-term outcomes.

## A gap in informal support

Some women experienced limited family support, mirroring findings from Tanzania, where OF patients reported lower social support compared to other gynecologic patients [57]. Previous studies underscored the significance of informal support for women facing reproductive health challenges, such as infertility and stillbirth, positively impacting their mental health outcomes [58–60]. In our study, participants noted the value of informal family support, which motivated them and eased their burdens post-OF surgery. Other research also highlights the role of informal support in women's reintegration after OF treatment, enhancing their emotional well-being [61, 62]. Initiating improved social support at fistula treatment facilities before discharge could involve healthcare professionals educating women and contacting family members to emphasize their support's importance [63]. Additionally, involving social workers and community-based rehabilitation workers in the transition process could help women access community resources [64]. Local NGOs focusing on OF could advocate for enhanced social support for these women.

## Limitations of the study

The current study has several limitations. Due to the topic's sensitivity, participants chose interview locations, often preferring public spaces like churches or health institutions. Home interviewees may have been inhibited for fear of being overheard, though precautions were taken to ensure privacy.

The study was conducted by a female Ph.D. student from an urban area; however, participants were rural women, potentially creating a power imbalance. To mitigate this, the investigator dressed and spoke in a rural style. Findings were translated from Amharic to English, risking some loss of meaning, but initial analysis was done in Amharic, and a professional translated the themes. Amharic words with English explanations were used where nuance was difficult to capture. The study's findings, based on the views of 21 Ethiopian women, may have limited transferability beyond this context.

## Conclusion

Our findings show that while OF surgery positively impacted women's lives by restoring physical well-being, it did not fully address their psychosocial and economic needs. Even after successful surgery, women often faced social challenges like economic hardship, reproductive health issues, and marital problems, which affected their social inclusion. These issues were more pronounced for those not physically cured. To address these challenges, rehabilitation

services and community-based programs are needed. The study suggests that obstetric fistula care should extend beyond surgery to include continuous community-based support focusing on women's sexual and psychological health and social participation.

## Acknowledgments

We thank Queen's University and the University of Gondar for granting ethical approval for our study. Special thanks to the Bahirdar Hamlin Fistula Treatment Facility and the University of Gondar Treatment Facility for facilitating participant access. We would also like to acknowledge the research assistants who helped locate participants and an English language consultant.

## Author Contributions

**Conceptualization:** Tibeb Zena Debele, Beata Batorowicz.

**Formal analysis:** Tibeb Zena Debele, Danielle Macdonald, Heather M. Aldersey, Zelalem Mengistu, Dawit Gebeyehu Mekonnen, Beata Batorowicz.

**Methodology:** Tibeb Zena Debele, Danielle Macdonald, Heather M. Aldersey, Zelalem Mengistu, Dawit Gebeyehu Mekonnen, Beata Batorowicz.

**Project administration:** Tibeb Zena Debele, Beata Batorowicz.

**Resources:** Tibeb Zena Debele.

**Supervision:** Tibeb Zena Debele, Beata Batorowicz.

**Validation:** Tibeb Zena Debele, Danielle Macdonald, Heather M. Aldersey, Beata Batorowicz.

**Visualization:** Tibeb Zena Debele, Danielle Macdonald, Heather M. Aldersey, Zelalem Mengistu, Beata Batorowicz.

**Writing – original draft:** Tibeb Zena Debele, Danielle Macdonald, Heather M. Aldersey, Zelalem Mengistu, Beata Batorowicz.

**Writing – review & editing:** Tibeb Zena Debele, Danielle Macdonald, Heather M. Aldersey, Zelalem Mengistu, Beata Batorowicz.

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
