## [Decision Letter · Decision Letter 0]

19 Apr 2024

PONE-D-23-41017“I became a person again”: Social Inclusion and participation experiences of Ethiopian women post-Obstetric Fistula surgical repair.PLOS ONE

Dear Dr. Debele,

Thank you for submitting your manuscript to PLOS ONE. After careful consideration, we feel that it has merit but does not fully meet PLOS ONE’s publication criteria as it currently stands. Therefore, we invite you to submit a revised version of the manuscript that addresses the points raised during the review process.

We look forward to receiving your revised manuscript.

Kind regards,

Sidrah Nausheen, FCPS

Academic Editor

PLOS ONE

Journal Requirements:

2. Thank you for stating the following financial disclosure: "Tibeb Debele received funding from the Mastercard Foundation and Queen's University dean's travel grant. "  

Reviewers' comments:

Reviewer's Responses to Questions

**Comments to the Author**

1. Is the manuscript technically sound, and do the data support the conclusions?

Reviewer #1: Yes

Reviewer #2: Yes

2. Has the statistical analysis been performed appropriately and rigorously? 

Reviewer #1: Yes

Reviewer #2: Yes

3. Have the authors made all data underlying the findings in their manuscript fully available?

Reviewer #1: Yes

Reviewer #2: Yes

4. Is the manuscript presented in an intelligible fashion and written in standard English?

Reviewer #1: Yes

Reviewer #2: Yes

5. Review Comments to the Author

Reviewer #1: Fistula skill are required to fix this problem of women. In developing countries fistula still remains the mainstay of problem, more of such publications are required in big centers like Ethiopia to highlight the impact on quality of life.

Reviewer #2: This is a comprehensive exploration of the lived experiences of women with Obstetric Fistula (OF), covering almost all relevant aspects of social participation and inclusive experiences. The inclusion of actual words and statements from the patients has supported the confirmability.

However, there are some areas that require consideration:

1.The manuscript is excessively lengthy (50 pages), which diverts readers' attention from the key messages. Adjusting the word count and reducing repetition of concepts and content would improve readability. Additionally, the format should align more closely with that of an article rather than a thesis.

2.The content of the tables shows duplication of the term "vesicovaginal fistula," which appears to be a typo error. Consistency in the number of digits after the decimal point should be ensured throughout, and percentages should sum up to 100 for the presented data.

3.The document requires a thorough review for minor language flow and grammatical errors.

4.Details are missing regarding how privacy was ensured during interviews conducted in various locations, such as churches and homes.

5.Additionally, information on the data collection process, interview timings, and participant selection criteria would enhance transparency and reproducibility.

6.What measures were taken to ensure reflexivity and transparency regarding the researcher's biases and assumptions should be elaborated upon.

7.Evidence of data saturation should be provided to ensure completeness of data collection.

8.Differences in experiences and inclusiveness based on factors such as the type of fistula, duration since surgery, and the extent of incontinence or number of surgeries is lacking.

9.The study's transferability aspect is commendable, with rich descriptions of findings. however, a discussion of the study's limitations and constraints affecting transferability needs consideration.

10.In conclusion section, comments on the implications of such an extensive review and potential policy changes are missing.

6. PLOS authors have the option to publish the peer review history of their article (what does this mean?). If published, this will include your full peer review and any attached files.

Reviewer #1: No

Reviewer #2: No

---

## [Author Response · Author response to Decision Letter 0]

23 May 2024

Dear Editor, 

Thank you very much. I have tried to incorporate all of the questions and feedbacks of the reviewers in the point by point response. For your questions, I have addressed all the comments given. One of the questions was regarding the roles of funders. I confirm that, The funders had no role in study design, data collection and analysis, decision to publish, or preparation of the manuscript.

---

## [Editor Report · Decision Letter 1]

31 May 2024

PONE-D-23-41017R1“I became a person again”: Social Inclusion and participation experiences of Ethiopian women post-Obstetric Fistula surgical repair.PLOS ONE

Dear Dr. Debele,

Thank you for submitting your manuscript to PLOS ONE. After careful consideration, we feel that it has merit but does not fully meet PLOS ONE’s publication criteria as it currently stands. Therefore, we invite you to submit a revised version of the manuscript that addresses the points raised during the review process.

the manuscript is written on very important topic and is generalizable, but it is too lengthy. kindly concise it by reducing repetition of words and sentences. correct grammatical mistakes   

Please submit your revised manuscript by Jul 15 2024 11:59PM If you will need more time than this to complete your revisions, please reply to this message or contact the journal office at plosone@plos.org. Please include the following items when submitting your revised manuscript:A rebuttal letter that responds to each point raised by the academic editor and reviewer(s). You should upload this letter as a separate file labeled 'Response to Reviewers'.A marked-up copy of your manuscript that highlights changes made to the original version. You should upload this as a separate file labeled 'Revised Manuscript with Track Changes'.An unmarked version of your revised paper without tracked changes. You should upload this as a separate file labeled 'Manuscript'.If applicable, we recommend that you deposit your laboratory protocols in protocols.io to enhance the reproducibility of your results. Protocols.io assigns your protocol its own identifier (DOI) so that it can be cited independently in the future. For instructions see: https://journals.plos.org/plosone/s/submission-guidelines#loc-laboratory-protocols. Additionally, PLOS ONE offers an option for publishing peer-reviewed Lab Protocol articles, which describe protocols hosted on protocols.io. Read more information on sharing protocols at https://plos.org/protocols?utm_medium=editorial-email&utm_source=authorletters&utm_campaign=protocols.

We look forward to receiving your revised manuscript.

Kind regards,

Sidrah Nausheen, FCPS

Academic Editor

PLOS ONE
---

## [Author Response · Author response to Decision Letter 1]

21 Jun 2024

I have addressed the reviewers comments to reduce the length of the manuscript

---

## [Editor Report · Decision Letter 2]

28 Jun 2024

“I became a person again”: Social Inclusion and participation experiences of Ethiopian women post-Obstetric Fistula surgical repair.

PONE-D-23-41017R2

Dear  Tibeb Zena Debele,

We’re pleased to inform you that your manuscript has been judged scientifically suitable for publication and will be formally accepted for publication once it meets all outstanding technical requirements.

Kind regards,

Sidrah Nausheen, FCPS

Academic Editor

PLOS ONE
---

## [Editor Report · Acceptance letter]

2 Jul 2024

PONE-D-23-41017R2 

PLOS ONE

Dear Dr. Debele, 

I'm pleased to inform you that your manuscript has been deemed suitable for publication in PLOS ONE. Congratulations! Your manuscript is now being handed over to our production team.

Kind regards, 

on behalf of

Dr. Sidrah Nausheen 

Academic Editor

PLOS ONE